# Gender Differences in Uptake, Adherence and Experiences: A Longitudinal, Mixed-Methods Study of a Physical Activity Referral Scheme in Scotland, UK

**DOI:** 10.3390/ijerph18041700

**Published:** 2021-02-10

**Authors:** Coral L. Hanson, Lis Neubeck, Richard G. Kyle, Norrie Brown, Robyn Gallagher, Robyn A. Clark, Sheona McHale, Susan Dawkes

**Affiliations:** 1School of Health and Social Care, Edinburgh Napier University, Sighthill Campus, Edinburgh EH11 4DN, UK; l.neubeck@napier.ac.uk (L.N.); n.brown@napier.ac.uk (N.B.); s.mchale@napier.ac.uk (S.M.); s.dawkes@rgu.ac.uk (S.D.); 2Sydney Nursing School, Charles Perkins Centre, Johns Hopkins Road, University of Sydney, Sydney, NSW 2006, Australia; robyn.gallagher@sydney.edu.au; 3Research & Evaluation Division, Knowledge Directorate, Public Health Wales, Cardiff CF10 4BZ, UK; richard.kyle@wales.nhs.uk; 4College of Nursing and Health Sciences, Flinders University, Adelaide, SA 5001, Australia; Robyn.clark@flinders.edu.au; 5School of Nursing, Midwifery and Paramedic Practice, Garthdee Campus, Robert Gordon University, Aberdeen AB10 7QE, UK

**Keywords:** Physical activity, public health, adherence, uptake, exercise referral, gender, mixed methods

## Abstract

Physical activity referral schemes (PARS) are implemented internationally to increase physical activity (PA), but evidence of effectiveness for population subgroups is equivocal. We examined gender differences for a Scottish PARS. This mixed-methods, concurrent longitudinal study had equal status quantitative and qualitative components. We conducted 348 telephone interviews across three time points (pre-scheme, 12 and 52 weeks). These included validated self-reported PA and exercise self-efficacy measures and open-ended questions about experiences. We recruited 136 participants, of whom 120 completed 12-week and 92 completed 52-week interviews. PARS uptake was 83.8% (114/136), and 12-week adherence for those who started was 43.0% (49/114). Living in less deprived areas was associated with better uptake (*p* = 0.021) and 12-week adherence (*p* = 0.020), and with male uptake (*p* = 0.024) in gender-stratified analysis. Female adherers significantly increased self-reported PA at 12 weeks (*p* = 0.005) but not 52 weeks. Males significantly increased exercise self-efficacy between baseline and 52 weeks (*p* = 0.009). Three qualitative themes and eight subthemes developed; gender perspectives, personal factors (health, social circumstances, transport and attendance benefits) and scheme factors (communication, social/staff support, individualisation and age appropriateness). Both genders valued the PARS. To increase uptake, adherence and PA, PARS should ensure timely, personalised communication, individualised, affordable PA and include mechanisms to re-engage those who disengage temporarily.

## 1. Introduction

Regular physical activity (PA) has a beneficial effect on the risk of cardiovascular disease (CVD), diabetes, some cancers and mortality from all causes [1]. Despite this, globally in 2018, one in four people undertook less than the 150 min of moderate PA per week recommended at this time [2]. In Scotland in the same period, 30% of males and 40% of females were insufficiently active and 21% of adults undertook less than 30 min of moderate PA per week [3]. Interventions are required to support those who are least active to increase PA and improve their health outcomes.

Physical activity referral schemes (PARS) are an internationally widespread intervention aimed at increasing PA [4,5,6]. In some countries, schemes focus on PA “prescription” by healthcare professionals (HCPs) [5,7]. However, in the United Kingdom (UK), PARS mainly involve HCPs referring patients with chronic or non-communicable diseases to leisure providers. These schemes include supervised PA programmes following an individualised assessment [8,9] and are usually 10–16 weeks in duration [10]. Evidence of long-term effectiveness for PARS in increasing PA is lacking [11,12]. Effectiveness appears to be influenced by who is referred (referral), how many of those referred initially participate (uptake) and, of those who start, how many continue to participate (adherence) [13]. There is a lack of understanding about whom PARS successfully engage with, and analyses are limited by a lack of standardised definitions for uptake and adherence [14]. These issues limit the ability of schemes to either inform HCPs about which of their patients are more likely to engage, or to develop and test new ways to engage with those that PARS do not currently reach successfully [15].

Studies examining associations between demographics and referral, uptake and adherence have reported equivocal results for PARS. A consistent finding is that males are less likely than females to be referred (33–41% male referral compared to 59–67% female referral) [16,17,18,19,20,21,22,23,24]. Male uptake has also been reported to be lower than female uptake in some studies [16,17,25], but others reported no association [19,26,27]. Similarly, studies reporting associations between gender and adherence are not conclusive, with some reporting that males are more likely to adhere [18,19,20], and others reporting no association [16,17]. We identified only one qualitative study reporting gendered barriers and enablers to PARS [28]. In this study, males considered being seen as active to be important to gender identity but lacked confidence and felt intimidated by younger fitter people [28]. This limited qualitative evidence gives only a superficial insight into gender differences within PARS.

A 2018 audit of Scottish PARS provision reported the existence of 26 schemes, 17 spanning local authority areas, 6 spanning National Health Service (NHS) Scotland regional health boards and 3 smaller schemes. The majority (53%) reported engaging with at least 60% of primary care practices in their area [29]. This audit did not report associations between demographics and engagement. Despite widespread provision, there is currently no national guidance for the implementation of PARS in Scotland, and we identified only two studies examining Scottish PARS [30,31], neither of which explored associations with gender.

Given that PARS are widely available in Scotland [29], there is a need to evaluate existing schemes to assess engagement and whether they support participants to increase PA. In addition, the equivocal evidence regarding differences in uptake and adherence between males and females means that there is a clear need to understand gender influences better. The lack of consistency in gender-based results in previous quantitative studies and the lack of qualitative exploration about gendered participation in PARS suggests that a mixed-methods approach is appropriate to increase understanding. This study therefore aimed to examine gender differences in uptake and adherence to a Scottish PARS and explore gendered perceptions of PARS experiences.

## 2. Materials and Methods

The study was a mixed-methods, concurrent longitudinal design with equal status quantitative and qualitative elements [32]. We conducted telephone interviews at three time points (pre-scheme, and after 12 and 52 weeks) that included validated quantitative questionnaires that have been previously used in PARS and/or other PA research (Global Physical Activity Questionnaire (GPAQ) [33] and Self-Efficacy and Exercise Habits Survey [34]), and open-ended questions about PARS experiences. The NHS Research and Ethics Committee (REF: 17/NI/0112), and Edinburgh Napier University School of Health and Social Care Research and Integrity Committee (REF: FHLSS/1856) approved the study.

### 2.1. Scheme Context

The PARS studied covered one of Scotland’s 32 unitary council areas, with a co-terminus NHS regional health board. Referrals were from HCPs in primary and secondary care to 11 local authority leisure centres and 2 community facilities. The primary aim of the PARS was to provide opportunities for adults with a range of medical conditions (CVD, history of or at risk of falls, diabetes, stroke, multiple sclerosis, chronic obstructive pulmonary disease (COPD), rheumatic disease and obesity) to take part in PA. The scheme received approximately 1000 referrals per year and was time-unlimited. Referrals were made via email on a standardised referral form and shared according to Caldicott Guidelines [35]. Patients gave signed consent for the transfer of information at referral. The scheme consisted of a range of functionally stratified group PA sessions (Table 1). HCPs assessed patients’ level of functionality at the point of referral.

Activity options included once a week attendance at the gym, seated exercise classes, circuit classes or aqua-based activities.

### 2.2. Sample

We recruited study participants over 11-months (March 2018–January 2019). We approached all HCPs (*n* = 58) who had referred ≥4 patients to the scheme in the previous year and asked them to inform patients about the study during referral. All those referred were eligible to take part, but we asked referrers to recruit both males and females to allow exploration of gender differences in outcomes and perspectives of PARS experience. HCPs sought written consent to share personal contact information (name, address and telephone number) with the study team. Researchers then recruited participants by attempting up to 10 contacts by telephone/text. Participants gave written signed consent to take part in the study, which included consent for the PARS provider to release data about the first 12 weeks of attendance. Participants also gave renewed verbal consent before each interview. Recruiting HCPs received monthly feedback on recruitment progress.

### 2.3. Data Collection

Researchers telephoned participants and worked through a single questionnaire at baseline. This contained validated instruments to measure PA (GPAQ) [33] and exercise self-efficacy (Self-Efficacy and Exercise Habits Survey) [34]. It also contained demographic/personal information (gender, age, postcode, employment status, education level, reason for referral, co-morbidities and intended mode of travel to the PARS venue) and open-ended questions about expectations and perceived influences on attendance. At 12 and 52 weeks, participants completed the validated questionnaires, discussed attendance and answered open-ended questions about PARS experiences. One researcher (CH) completed all pre-scheme and 12-week interviews, and two researchers (CH and SM) completed 52-week interviews. We recorded responses to quantitative elements on paper-based questionnaires and transferred them to a Microsoft Excel (Microsoft Corporation, St. Redmond, WA, USA) spreadsheet following each interview. The PARS provider supplied researchers with individual attendance data for the first 12 weeks of participation for all study participants.

Researchers did not audio-record the interviews due to the balance of closed versus open questions but instead made extensive field notes about qualitative responses to open-ended questions during and after interviews. These included verbatim quotes to highlight participant experiences, and interviewer perceptions about the interaction between scheme function and participant experiences. Any identifiable elements, such as names or venues attended, were excluded from field notes to ensure anonymity. All participants were assigned a numeric study ID.

### 2.4. Quantitative Analysis

We used SPSS V26 (IBM, New York, NY, USA) to analyse quantitative data. Descriptive statistical analysis summarised participant characteristics (gender, age group, education level, employment status, socioeconomic status using Scottish Index of Multiple Deprivation (SIMD) quintiles calculated from postcodes, reason for referral, number of co-morbidities and transport). We classified participants by engagement level during the first 12 weeks of the study; (1) non-starter (did not attend any sessions), (2) dropout (attended < 67%, 1–7 sessions), or (3) adherer (attended ≥ 67%, ≥8 sessions). The dropout/adherence cut point was determined by the protocol of the PARS studied. We combined classifications 2 and 3 to measure uptake and used classification 3 to measure adherence. We used univariate analyses to examine whether demographics (gender, age and SIMD quintile) and referral factors (reason for referral, number of co-morbidities and transport) predicted uptake and adherence. Where cell size was less than five, we aggregated data. Analyses (Pearson’s χ^2^ and Mann–Whitney U tests) examined whether there were significant associations (*p* < 0.05 with 95% confidence intervals) for these characteristics between (i) non-starters and starters and (ii) dropouts and adherers. Total weekly activity (MET-min/week) and sedentary time (min/day) were calculated from GPAQ data [33], and making time for PA and sticking to PA scores were calculated from the Self-efficacy and Exercise Habits Survey [34]. We examined data distribution and used repeated measures analyses to examine changes after 12 and 52 weeks compared to baseline. Where differences were apparent, we examined these using post hoc tests with a Bonferroni correction.

### 2.5. Qualitative Analysis

During week 12 and 52 interviews, we checked the accuracy of field notes by reflecting them back to participants during conversations, making corrections where necessary. For example, we reminded a participant that during the first interview, they mentioned that they felt worried about attending the PARS. This allowed the participant to confirm, or correct, this point and encouraged them to expand further if they wanted to. We analysed field notes using NVIVO V12 (QSR International, Melbourne, Victoria, Australia). Data were subject to thematic analysis [36] using the framework approach [37]. After reading and rereading the field notes to become familiar with the data, CH (who had seven years’ experience of qualitative research) created open nodes (*n* = 49) recording preliminary concepts for 10 participants. CH presented the initial codes at a data workshop involving five authors (CH, LN, SD, NB, RGK), who had independently read the ten sets of field notes. After critical discussion, we developed an initial framework containing six major themes (gender, PA, scheme process factors, personal factors, “it’s not for me” and benefits of attendance) and 11 subthemes (previous PA, current PA, communication, scheme sessions, transport and accessibility, staff support, social support, age, individualisation, health and confidence). After analysing ten more sets of field notes, we refined the framework and completed analysis of all cases. We explored connections within and between participants and categories. During the final analysis, we developed three overarching themes and eight subthemes giving insight into participant experiences.

### 2.6. Data Integration

Preliminary analysis of both quantitative and qualitative data took place at several intermediate time points in the study (after 6 and 12 months). Progress was reported during quarterly steering group meetings and the integration of findings discussed by all authors. Final integration took place after separate full analysis of results for quantitative and qualitative elements with equal status given to both elements.

## 3. Results

We received participant details for 209 referrals and recruited 136 people to the study (Figure 1).

### 3.1. Quantitative Findings

#### 3.1.1. Participant Characteristics

Participants were 52.2% male, older (67.6% ≥60 years), from the least two deprived SIMD quintiles (55.9%) and had at least three co-morbidities (51.5%) (Table 2). The median age of participants was 67.0 (IQR 55.0–73.7) years. The median number of co-morbidities was significantly different between study dropouts and study completers (U = 2445, z = 2.012, *p* = 0.044). Median number of co-morbidities for study dropouts was 2.0 (IQR 1.0–3.8) and for study completers was 3.0 (IQR 2.0–4.0).

Males were more likely to be referred for CVD secondary prevention than females (39.4% vs 10.8%) (Table 3). There was a significant association between gender and reason for referral at baseline (χ^2^ (3) = 18.34, *p* < 0.001, φc = 0.367). This association remained for 12-week study completers (χ^2^ (3) = 17.18, *p* < 0.001, φc = 0.378) and 52-week study completers (χ^2^ (3) = 12.35, *p* < 0.006, φc = 0.366).

#### 3.1.2. Attendance, Drop out and Adherence

The median time from referral to first attendance was 44.0 days (IQR 31.8–66.0) (males, 43 days, IQR 28.0–62.5 and females 45 days, IQR 34.5–69.5). Based on actual attendance data provided by the PARS for all study participants (*n* = 136), 114 (83.8%) attended at least one PARS session (Table 4).

Living in less deprived SIMD quintiles was significantly associated with uptake (χ^2^ (4) = 11.6, *p* = 0.021, φc = 0.353) and 12-week adherence (attending ≥ 8 sessions during the first 12 weeks) for those who started (χ^2^ (4) = 11.6, *p* = 0.020, φc = 0.377). The association remained for male uptake during gender-stratified analyses (χ^2^ (4) = 11.3, *p* = 0.024, φc = 0.398).

Of the 120 respondents at 12 weeks, 22 (18.3%) had not attended, 98 (81.7%) had started attending, 41 (34.2%) dropped out after attending 1–7 sessions and 57 (47.5%) attended ≥ 8 sessions. Of the 92 respondents at 52 weeks, only 1 person who had not started at 12 weeks began to attend after this period. Of the study completers, 34 (37.0%) were still classified as adherers at 52 weeks (Table 5). There were no significant demographic or personal characteristic differences between adherers and dropouts at 12 or 52 weeks for study completers.

#### 3.1.3. Behaviour Change for Study Completers

Female adherers significantly increased PA between baseline and 12 weeks (*p* = 0.005) but did not maintain this at 52 weeks. Those who did not start the scheme significantly increased PA between baseline and 52 weeks (*p* = 0.014). Male adherers decreased their sedentary behaviour between 12 and 52 weeks (*p* = 0.004) (Table 6).

Male adherers significantly increased “making time for PA” scores between baseline and 52 weeks (*p* < 0.0005) and “sticking to PA” scores between baseline and 52 weeks (*p* = 0.004) and between 12 and 52 weeks (*p* = 0.005) (Table 7).

### 3.2. Qualitative Analysis

We conducted 348 interviews. Median interview time was 44 min (range 22–73 min). Three overarching themes and eight subthemes influencing PARS participation developed: gender perspectives, personal factors and scheme process factors. Gender balance did not affect participation, but innate gender bias (e.g., males attended to work, while females socialised) was evident when discussing scheme suitability. Personal factors included previous PA, health, social circumstances, transport and the psychological and physical benefits of attending. Scheme process factors included communication, social and staff support and a lack of individualisation/perceptions about age appropriateness, leading some participants to conclude that the scheme “was not for me”. It was clear that quantitative classification of participants as dropouts or adherers was too simplistic, as some who had attended less than eight sessions considered themselves to still be attending and vice versa. To illustrate this, we added quantitative adherence status to all participant quotes.

#### 3.2.1. Gender Perspectives

Participants reported sessions were often a 50/50 gender split, although some exercise classes had a higher proportion of females. Gender variation in sessions did not influence attendance decisions. Only rarely did participants suggest they would prefer single-gender sessions. In exception, one man, who attended the gym during times with shared public use, described how “*the women in the gym tend to be a very fit and intimidating. I am not intimidated by butch men—I don’t want to be like that*” (Participant 132, male, <59 years, 12-week dropout).

Gender comparisons were more likely to influence perceptions of scheme suitability for males. Male participants often focused on physical fitness gains and some considered that the scheme was more about the social aspect than having a workout. Participants expressed views such as “*most of the women (and some men) are there to exercise their jaws more than their body*” (participant 67, male, <59 years, 12-week dropout) and “*some of the women were there to compare notes and have a chat … the men were there to do work*” (participant 102, male, 70+ years, 12-week adherer).

Females tended to emphasise enjoyment and the social aspect as important influences on continued attendance,


*“It is lots of fun and there are nice people there. The main thing has been meeting other people and getting out of the house. This has made me feel a lot happier. I just chat on with whoever happens to be there”*
(Participant 109, female, 60–69 years, 12-week dropout)

Although such statements appeared to confirm male perceptions about female attendance reasons, females also wanted to improve fitness and health; “*the sessions are social but exercise is the main reason to attend*” (Participant 43, female, 70+ years, 12-week adherer). Females were more overt about comparing their own health and abilities with other attendees, and this encouraged attendance for some: “*there are people with back problems, diabetes, stroke and multiple sclerosis. The youngest is about 40 and the oldest 84. When I look around, I think ‘how lucky am I’?*” (Participant 62, female, 60–69 years, 12-week dropout).

Some males expressed a preference to include a competitive element, which they felt existed subliminally anyway. This led to suggestions for sporting activities in the PARS: “*something like walking football would be good—where the activity introduces competition, but ability is less of a problem*” (Participant 118, male, <59 years, 12-week dropout). No females mentioned a desire for competition within sessions.

#### 3.2.2. Personal Factors

Participants identified personal factors that influenced attendance; previous PA, health, social circumstances, transport and the benefits resulting from attendance, e.g., improved confidence, mental and physical health.

#### Previous Physical Activity Levels

Many participants reported previously being active. Some reflected back to active younger years and described gradual reductions in PA with increasing age. Others had been active more recently, but either an injury—“*I used to be very active but have struggled since having knee surgery*” (Participant 35, female, 60–69 years, 12-week adherer)—or an acute medical issue—“*I was previously very active. I was clearing my elderly neighbour’s path of snow and had a heart attack*” (Participant 42, female, 60–69 years, 12-week dropout)—now prevented participation. During baseline interviews, several participants described being worried about exercising, as they were unsure whether it would make their condition worse or cause another injury/fall. Prior to attendance at the PARS, the most commonly reported activities were walking or having recently completed clinically based rehabilitation programmes. Those who had attended clinical rehabilitation were more confident about starting the PARS, for example talking about having “*completed cardiac rehabilitation and I am ready to move on*” (Participant 42, female, 60–69 years, 12-week dropout).

#### Health

Participants described how poor health prevented uptake for some and was a short or long-term barrier to attendance for others. Many had complex health problems with multiple co-morbidities. Health issues reported as occurring during the study included viruses, falls, back pain, routine surgery, temporary worsening of established conditions (e.g., chest infections for those with COPD) and diagnosis of new conditions. For some, attending the PARS had worsened existing problems, e.g., “*my hip was hurting and the chair was not comfortable*” (Participant 83, female, 60–69 years, 12-week dropout), causing dropout. For others, short-term illness led to lower attendance. “*I haven’t been for the last four weeks because of a virus. I am feeling better now though and intending to go back this week*” (Participant 90, male, aged 70+ years, 12-week adherer).

During 52-week interviews, some participants reported that their conditions had worsened. As at 12 weeks, there were short-term—“*I have an issue with sciatica which is limiting my PA levels, need to get back but need these issues addressed first*” (Participant 39, female, <59 years, 12-week dropout) and long-term—“*I haven’t been back. My asthma has been really bad and I am struggling. Walking is difficult due to my back problems*” (Participant 88, female, aged 70+, 12-week adherer) issues preventing a return to sessions.

#### Social Circumstances

Participants often discussed complex home lives and described how their social circumstances affected their choices about PA either positively: “*my son lives locally and give me lots of support to exercise*” (Participant 31, female, 70+ years, 12-week dropout); or negatively: “*it makes me very stressed because my husband does not support me trying to make lifestyle changes*” (Participant 11, female, <59 years, 12-week adherer). Where partners and/or friends did not take part in PA, it was more difficult for participants to make positive changes.

Some participants reported that they were unable to attend due to caring responsibilities. Looking after children/grandchildren during school holidays or while sick provided a temporary attendance barrier for some, while caring responsibilities for relatives provided a long-term barrier for others. One participant, who had attended 10 sessions, stated: “*It was a good session, but I stopped going. I had to prioritise my husband. He went into hospital for a femoral bypass and I have been too busy looking after him to attend*” (Participant 2, female, 70+ years, 12-week adherer). At 52 weeks, this participant’s husband now had terminal cancer. She was still positive about her time attending the PARS and reported doing some home exercises but was unable to return due to caring responsibilities.

Attendance was more challenging for those who worked. For some, shift patterns made consistent attendance difficult, while for those working full-time office hours, PARS attendance was impossible due to the predominance of daytime sessions. Even for those willing to take time out of their working day, “*there is not enough time at lunchtime, as the travel to and from the centre make it much more than an hour*” (participant 128, female, <59 years, 12-week dropout).

#### Transport

The majority of participants reported that they used their own transport, or that a friend/family member gave them a lift to the PARS. Older females tended to be more reliant on others for transport. At some leisure sites, busy car parks and limited disabled parking bays meant parking could be an issue. Where participants used public transport, they described inconvenient routes involving several buses and arrival/departure times incompatible with sessions. A free “dial-a-ride” service was described as “*really convenient*” with “*a total transport and wait time of approximately 30 min each journey*” (Participant 7, female, 70+ years, 12-week dropout). Attending sessions that finished at 3 p.m. was problematic for those using this service, however, as it also served as school transport and was unavailable for return home journeys.

#### Physical and Mental Benefits of Attendance

Many participants stated that they enjoyed attending. The least active referrals reported that the scheme helped to improve confidence, functional fitness (such as ability to dress without help or walk up stairs), balance and strength: “*I have thrown away my walking stick and now feel more confident about walking*” (Participant 25, male, 70+ years, 12-week dropout). The scheme also encouraged some to learn about safe and appropriate PA: “*I have learned that it is good to get out of breath and how to pace myself*” (Participant 1, female, 60–69 years, 12-week adherer). In addition to physical benefits, the PARS improved mental health making participants “*feel happier and more positive*” (Participant 40, female, <59 years, 12-week adherer). Mental health benefits were often unexpected by participants.

#### 3.2.3. Scheme Process Factors

Scheme factors, such as communication and social/staff support influenced attendance. Some participants stated that the PARS was “*not for me*” (Participant 133, male, 60–69, 12-week dropout, Participant 136, female, <59 years, 12-week dropout). Exploration of this statement revealed a range of issues. Dropouts considered that the scheme was not age-appropriate, lacked individualisation, had inappropriate levels of PA and was not enjoyable.

#### Communication with the PARS

Participants reported some issues with pre-scheme communication, and this was the main reason given for not starting. Some said that they had not been contacted by the PARS, while others were “*still waiting to hear from the scheme after an initial letter I had saying that I was on a waiting list*” (Participant 48, male, 60–69 years, non-starter). Some participants did receive an invitation to attend but did not go to their allotted session because “*the timing didn’t fit with my shift pattern. I didn’t see the point in going as I wouldn’t be able to keep attending*” (Participant 92, male, <59 years, non-starter). Non-starters and those who dropped out after only attending a few times received no further contact from the scheme. A few participants described how they were “*keeping in touch with the instructor. I have her mobile number so can either phone her or the sports centre to say when I want to go back*” (Participant 52, female, 70+ years, 12-week adherer). This was not consistent, with others reporting that when they stopped attending no one contacted them as it “*was not that sort of scheme*” (Participant 80, male, 70+ years, 12-week dropout).

#### Social and Staff Support

Social support influenced how much participants enjoyed the PARS and was one of the most important facilitators of adherence. Participants often reported that prior to starting, they felt “*a wee bit apprehensive but once I walked through the door everyone was really nice*” (Participant 61, male, 60–69 years, 12-week dropout). The exercise class format was particularly beneficial in promoting social interaction: “*I enjoyed meeting other people and the circuit means that you can mix with different people*” (Participant 7, female, 70+ years, 12-week adherer). Participants described the friendly nature of the groups, although many commented that interactions were limited to sessions rather than developing into friendships outside the PARS. Peer support also provided opportunities to discuss health problems: “*There are three people with multiple sclerosis—so it is really good to be able to open up and talk about it. This is the best bit of it*” (Participant 95, female, <59 years, 12-week dropout). Interaction with others not only increased enjoyment, it also encouraged attendance: “*other people in the session are expecting me to be there, and this gives me the motivation to attend*” (Participant 32, male, <59 years, 12-week adherer).

Those who adhered generally described staff in very positive terms using language such as “*understanding*” (Participant 3, female, 70+ years, 12-week dropout), “*encouraging*” (Participant 18, female, 70+ years, 12-week dropout), “*attentive*” (Participant 121, female, aged <59 years, 12-week adherer) and “*knowledgeable*” (Participant 76, male, <59 years, 12-week adherer). Staff provided guidance about safe and effective PA and encouraged further activity outside of sessions. There was a recognition that the number of participants in each session limited staff attention, and this discouraged attendance for some.

#### “It’s Not for Me”

Dropouts often described initial experiences using the phrase “it’s not for me”. Further exploration highlighted a range of scheme-related issues. The first was age appropriateness, with some participants under 50 reporting issues with social mix rather than inappropriate exercise intensity levels: “*I didn’t feel comfortable with all the pensioners and bumped into one of my teachers, which made me feel awkward*” (Participant 107, female, <59 years, 12-week dropout). Feeling that sessions were not age-appropriate did not necessarily prevent attendance: “*Most of the folk in the sessions are quite a bit older, so I feel that it is not really aimed at me. I am the youngest by about 10 years, but they are good people and we have a laugh*” (Participant 51, male, <59 years, 12-week dropout). However, age differences contributed to feelings of low self-esteem: “*my friends go to the gym and lift weights. I can’t do that and I don’t want to spread about that I am going to an oldies class*” (Participant 51, male < 59 years, 12-week dropout).

Participants reported that the scheme was not individualised, with some saying that sessions were too easy, calling into question the tiered class allocation based on ability. One participant, referred for weight management, described the session as “*so slow paced*” (Participant 116, female, <59 years, dropout). Some suggested that sessions were allocated based on PARS availability rather than individual requirements: “*there was a gap, so we will place you there, rather than finding out what I really wanted*” (Participant 74, male, 70+ years, non-starter). Participant 90 (70+ years, 12-week adherer), who reported a short-term barrier of ill health during his 12-week interview, returned to sessions, but during his 52-week interview reported that he was no longer attending because the class “*was no longer for me*”. He now considered it too easy, crowded and had limited equipment. Instead, he had joined a private gym and was attending four times per week.

## 4. Discussion

Both males and females valued the PARS. Gender was not associated with uptake or 12–week adherence but living in less deprived SIMD quintiles was associated with both. This association remained for male uptake during gender-stratified analyses. Females significantly increased self-reported short-term PA only. Males significantly increased exercise self-efficacy between baseline and 52 weeks, but this did not translate into significantly increased PA. Our qualitative analysis suggested that gender balance within PARS sessions did not influence attendance. Instead, the most influential factors determining adherence or dropout were personal factors (health, social circumstances, transport and the psychological and physical benefits of attending) and scheme process factors (communication, social/staff support, individualisation and age appropriateness). These findings suggest that regardless of gender, PARS participants require more support to increase both short- and long-term PA levels. Furthermore, schemes need to address delivery process factors as an immediate priority and then reassess scheme success in light of improved delivery.

### 4.1. Uptake

Uptake in this study (83.3%) was slightly above the pooled uptake of 80% (95% CI 61% to 98%) for randomised controlled trials (RCTs) and observational studies of 28–81% reported in the most recent systematic review of effectiveness [12]. However, internal evaluation of the PARS in our study indicated a lower uptake of 58.4% [38], indicating a potential bias in intention to take part in the PARS for those who joined the study compared with those who did not (34.9% of potential participant details received from HCPs). The only demographic factor associated with uptake in this study was living in less deprived areas, in agreement with one previous study [19] but in contrast to others [16,17,26] that reported no association. We recommend that the PARS studied should engage with those referred from more deprived areas, particularly males in light of findings, to explore what extra support is required to encourage uptake.

Median time from referral to first attendance was 44.0 days (IQR 31.8–66.0). To our knowledge, this is the first study to report the time between referral and starting PARS, which may contribute to uptake decisions. Few PARS studies have engaged with non-starters to explore uptake, although we have previously reported contributing factors as poor health and social anxiety [39]. Furthermore, in exercise-based clinical rehabilitation studies, pre-programme contact and individualised information are important factors in engagement decisions [40,41,42]. Given the long wait time, limited pre-scheme contact and lack of individualisation reported in our study, we suggest PARS prioritise personal and timely initial contact, and the individualisation of attendance information. Additionally, PARS should assess functional ability via an individualised assessment prior to commencement to ensure that participants are offered appropriate PA options. This may improve perceived suitability and allow referrals to feel that activities “fit” their PA needs.

Systematic review evidence has highlighted that previously active PARS participants are more likely to increase PA as a result of PARS participation [12], and our previous qualitative work reported better engagement with this group [39]. What our study adds is that non-starters also reported being previously active. This indicates that HCPs may make decisions about who to refer partially based on whether they perceive patients are motivated to be active or not, as reported in previous studies exploring the promotion of PA by HCPs [43,44,45,46]. Further research is required to understand better how to engage those who have not been previously active to increase PA.

### 4.2. Adherence

Twelve-week adherence in this study was 43.0%, but as with uptake, internal evaluation by the PARS reported lower adherence (36.7%) [38]. This compares with a pooled adherence rate of 37% (95% CI 20–54%) from RCTs and of 12–70% in observational studies [12]. Comparison of adherence between studies is difficult as there is no standardised definition. Some studies report adherence as attendance at a final assessment [16,17,18] rather than at a number of PA sessions [19,24,31,47]. Where studies use actual attendance as an indicator of adherence, there is a variation in what percentage of possible attendances qualifies as adherence. Our findings suggest that quantitative analysis using a set number of sessions as an indicator of adherence is too simplistic as personal circumstances such as short-term illness or caring responsibilities led to temporary disengagement rather than dropout. PARS could improve adherence by creating formal processes, such as text-based contact systems (or telephone/postal systems for those who may be digitally excluded) that seek to re-engage those who stop attending regularly, particularly those who suffer from cyclical poor health, e.g., those with COPD having recurrent chest infections. Given the chronic medical conditions that participants had, temporary worsening of conditions was relatively commonplace, highlighting the need for contingency to enable these people to re-engage the PARS after short-term absence.

### 4.3. Change in Behaviour

In this study, only female participants significantly increased short-term PA, and this did not continue at 52 weeks. Systematic reviews report short-term increases in PA [11,12]. To our knowledge, gender differences in PA increases have not been evident in other studies, although significant mental health outcomes for females were reported by Murphy et al. [21]. We also report increased exercise self-efficacy for males, although this did not result in increased PA. One possible explanation is that males in our study were significantly more likely to be referred to the PARS for CVD (after completion of cardiac rehabilitation). It is possible that this group increased activity during earlier phases of rehabilitation and the PARS allowed for maintenance. Due to sample size, we were unable to perform statistical analysis to examine this. The PARS studied offered once per week sessions, which may have limited potential to increase PA. To encourage participants to achieve the 150 min of recommended PA [48], PARS must consider how to promote PA outside of scheme sessions.

### 4.4. Strengths and Limitations

Our study represented the naturalistic setting that PARS serve, which we consider essential to inform practice. We recruited participants at the point of referral, allowing for the inclusion of non-engagers in analyses. We included 52-week follow up, which is lacking in many PARS studies. The fully mixed-methods, concurrent design included a large qualitative sample, particularly given the longitudinal nature of the study.

We used the GPAQ to measure self-reported PA. The questionnaire has been extensively utilised and validated, but we recognise that objective monitoring is preferable for most accurate representation of activity. Given the longitudinal nature of this study and the geographical area covered, this was not feasible.

We did not audio-record interviews due to the balance of closed questions that required a tick box responses (where recording was not necessary) versus open questions that required a more in-depth recording of responses. Instead, we made detailed field notes (including verbatim quotes where possible) about open-ended responses. We acknowledge that some detail may have been lost, but this pragmatic approach allowed for the inclusion of more study participants.

## 5. Conclusions

We report short-term increases in PA for females resulting from PARS participation and significantly better uptake and adherence for those living in less deprived areas. Our qualitative analysis suggests that both genders valued the PARS. To improve uptake and adherence, PARS should ensure timely and personalised communication with referrals, that PA is individualised and affordable, and that mechanisms exist to engage with those who stop attending due to short-term issues such as illness or caring responsibilities.

## Figures and Tables

**Figure 1 ijerph-18-01700-f001:**
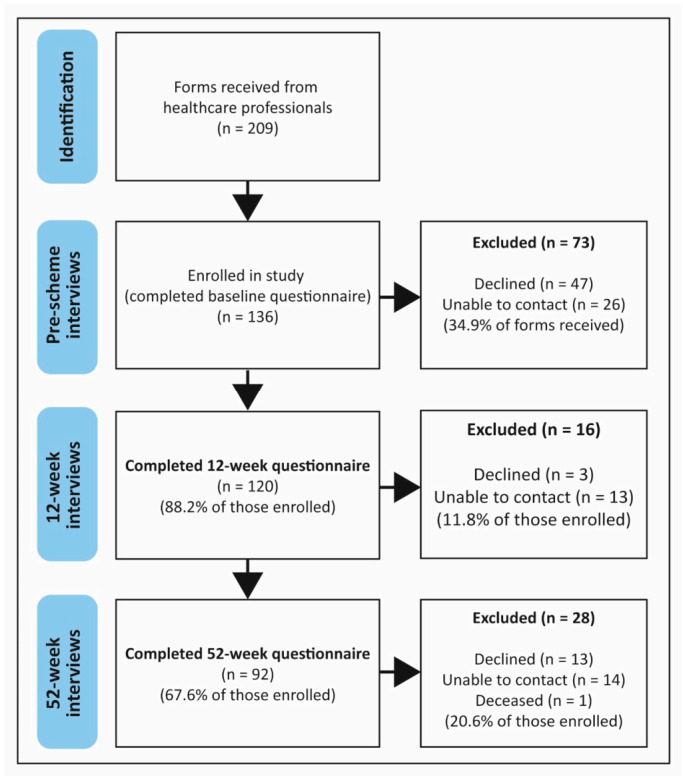
Study recruitment and participant flow.

**Table 1 ijerph-18-01700-t001:** Functionally stratified activity sessions.

Functional Level	Description
Level 1	Limited standing, balance and require mobility aid
Level 2	Mobile (without aid) but have difficulty with movement or activities of daily living
Level 3	Independently mobile
Level 4	Independently mobile and physically active

**Table 2 ijerph-18-01700-t002:** Participant characteristics at baseline, 12 and 52 weeks.

Characteristic	Baseline		12 Weeks		52 Weeks	
	*n* (136)	%	*n* (120)	%	*n* (92)	%
Gender						
Female	65	47.8	59	49.2	42	45.7
Male	71	52.2	61	50.8	50	54.3
Age group						
<50	21	15.4	15	12.5	7	7.6
50–59	23	16.9	20	16.7	17	18.5
60–69	40	29.4	37	30.8	32	34.8
70+	52	38.2	48	40.0	36	9.1
SIMD *						
1 (most deprived)	16	11.8	15	12.5	9	9.8
2	25	18.4	19	15.8	16	17.4
3	19	14.0	16	13.3	12	13.0
4	31	22.8	27	22.5	23	25.0
5 (least deprived)	45	33.1	43	35.8	32	34.8
Education Level						
Primary	8	5.9	7	5.8	5	5.4
Secondary	73	53.7	63	52.5	50	54.4
Post-secondary further education	18	13.2	23	19.2	16	17.4
Bachelor degree or higher	29	21.4	27	22.5	21	13.0
Employment status						
Employed/self-employed	28	20.6	26	21.7	19	20.7
Retired	84	61.8	76	63.3	60	65.2
Claiming incapacity benefit/other	24	17.6	18	15.0	13	14.1
Reason for referral						
COPD	15	11	14	11.7	9	9.8
CVD secondary prevention	35	25.7	33	27.5	30	32.6
Falls	16	11.8	13	10.8	9	9.8
Stroke	13	9.6	12	10.0	11	12.0
Type 1 or 2 diabetes	8	5.9	6	5.0	5	5.4
Obesity	29	21.3	25	20.8	17	18.5
Multiple sclerosis/rheumatic disease	20	14.7	17	14.2	11	11.9
No. of co-morbidities						
0–1	31	22.8	27	22.5	17	18.5
2	35	25.7	29	24.2	21	22.8
3+	70	51.5	64	53.3	54	58.7
Transport to sessions						
Private car	100	73.5	89	74.1	68	73.9
Public transport	19	14	16	13.3	13	14.1
Other (walking, cycling)	17	12.5	15	12.5	11	11.9
Type of session attended						
Circuit class	N/A		61	50.8	52	56.5
Gym			14	11.7	12	13
Gym circuit			11	9.2	8	8.7
Seated exercise/aqua aerobics			12	10.0	10	10.9
Did not attend any sessions			22	18.3	10	10.9

* Scottish Index of Multiple Deprivation Quintile.

**Table 3 ijerph-18-01700-t003:** Participant characteristics by gender at baseline, 12 and 52 weeks.

Characteristic	Baseline (*n* = 136)				12-Week Study Completers (*n* = 120)				52-Week Study Completers (*n* = 92)			
	Female		Male		Female		Male		Female		Male	
	*n* (65)	%	*n* (71)	%	*n* (59)	%	*n* (61)	%	*n* (42)	%	*n* (50)	%
Age group												
<59	24	36.3	20	28.2	19	32.2	16	26.2	10	23.8	14	28
60–69	19	29.2	21	29.6	18	30.5	19	31.1	17	40.5	15	30
70+	22	33.8	30	42.3	22	37.3	26	42.6	15	35.7	21	42
SIMD *												
1 (most deprived) and 2	22	33.8	19	26.8	18	30.5	16	26.2	10	23.8	15	30
3	8	12.3	11	15.5	7	11.9	9	14.8	6	14.3	6	12
4	13	20	18	25.4	12	20.3	15	24.6	10	23.8	13	26
5 (least deprived)	22	33.8	23	32.4	22	37.3	21	34.4	16	38.1	16	32
Education Level												
Primary/secondary/further education	50	76.9	57	80.3	44	75.6	49	80.3	30	71.4	41	82
Bachelor degree or higher	15	23.1	16	19.7	15	25.4	12	19.7	12	28.6	9	18
Employment status												
Employed/self-employed	12	18.5	16	22.5	11	18.6	15	24.6	5	11.9	14	28
Retired	39	60	45	63.4	38	64.4	38	62.3	30	71.4	30	60
Claiming incapacity benefit/other	14	21.5	10	14.1	10	16.9	8	13.1	7	16.7	6	12
Reason for referral												
CVD secondary prevention	7	10.8	28	39.4	7	11.9	26	42.6	7	16.7	23	46
COPD and falls prevention	19	29.2	12	16.9	18	30.5	9	14.8	13	31	5	10
Obesity	20	30.8	9	12.7	17	28.8	7	11.5	10	23.8	7	14
Other **	19	29.2	22	31	11	32.2	18	29.5	12	28.6	15	30
No. of co-morbidities												
0–1	14	21.5	17	23.9	14	23.7	13	21.3	7	16.7	10	20
2	17	26.2	18	25.4	13	22	16	26.2	7	16.7	14	28
3+	34	52.3	36	50.7	32	54.2	32	52.5	28	66.7	26	52
Transport to sessions												
Private car	68	73.9	53	74.6	43	74.1	46	75.4	30	71.4	38	77.6
Public transport, walking or cycling	24	26	16	22.4	15	25.9	12	19.7	12	28.6	11	22.4
Type of session attended												
Exercise class (circuit/seated/aqua)	N/A				37	62.8	36	59.1	31	73.8	31	62
Gym/gym circuit					10	16.9	15	24.6	6	14.3	14	28
Did not attend any sessions					12	20.3	10	16.4	5	11.9	5	10

* Scottish Index of Multiple Deprivation Quintile ** Multiple sclerosis, rheumatic disease, stroke, type 1 or 2 diabetes.

**Table 4 ijerph-18-01700-t004:** Uptake, dropout and adherence.

	All Participants		Females		Males	
	(*n* = 136)	%	(*n* = 65)	%	(*n* = 71)	%
Did not start	22	16.2	11	16.9	11	15.5
Uptake (attended at least one session)	114	83.8	54	83.1	60	84.5
Dropout (of those who started, those who attended 1–7 sessions)	65	57.0	30	55.6	35	58.3
Adherence (of those who started, those who attended ≥ 8 sessions)	49	43.0	24	44.4	25	41.7
Overall scheme adherence (of those referred, those who attended ≥ 8 sessions)	49	36.0	24	36.9	25	35.2

**Table 5 ijerph-18-01700-t005:** Study completers: adherence at 12 and 52 weeks.

Completers (*n* = 92) *	12 Weeks		52 Weeks	
	*n*	%	*n*	%
Did not start	11	12.2	10	10.9
Uptake (attended at least one session after referral)	81	88.0	82	89.1
Dropout (of those who started, those who attended 1–7 sessions in previous 12 weeks)	38	57.0	40	43.5
Adherence (of those who started, those who attended ≥ 8 sessions in previous 12 weeks)	43	53.1	34	41.5
Overall scheme adherence (of those referred, those who attended ≥ 8 sessions in previous 12 weeks)	43	46.7	34	37.0

* Unable to report gender differences due to cell sizes of <5.

**Table 6 ijerph-18-01700-t006:** Study completers: change in physical activity and sedentary behaviour.

Measure	Classification	Attendance Status	Baseline		12 Weeks		52 Weeks		Significance		
			Median	IQR	Median	IQR	Median	IQR	Overall α < 0.05	a α < 0.0167	b α < 0.0167
Global Physical Activity Questionnaire	Total weekly PA (MET-Min/Week)	All study completers (*n* = 92)	900.0	210.0–1980.0	960.0	480.0–2175.0	780.0	360.0–1440.0	0.057		
		Female study completers (*n* = 42)	520.00 ^a^	0.0–1140.0	760.0 ^a^	340.0–1800.0	620.0	240.0–1095.0	0.004	0.001	
		Male study completers (*n* = 50)	1440.0	465.0–2400.0	1220.0	590–2450.0	980.0	405.0–1710.0	0.214		
		12-week adherers (*n* = 43)	720.0	240.0–2040.0	960.0	400.0–2180.0	720.0	360.0–1440.0	0.105		
		Female 12-week adherers (19)	360.0 ^a^	0.0–960.0	720.0 ^a^	280.0–2160.0	440.0	240.0–720.0	0.008	0.005	
		Male 12-week adherers (*n* = 24)	1410.0	450.0–2370.0	1340.0	695.0–2250.0	960.0	435.0–1950.0	0.740		
		12-week dropouts (*n* = 38)	1200.0	230.0–2520.0	1000.0	240.0–2040.0	880.0	230.0–1500.0	0.055		
		Non-starter (*n* = 11)	480.0 ^a^	0.0–1440.0	880.0	623.0–1320.0	720.0 ^a^	560.0–1680.0	0.045	0.014	
	Sedentary behaviour (Min/day)	All study completers (*n* = 92)	540.0 ^a^	360.0–682.5	540.0 ^b^	420.0–660.0	480.0 ^a b^	300.0–600.0	<0.0005	0.002	0.001
		Female study completers (*n* = 42)	600.0	360.0–720.0	555.0	420.0–660.0	480.0	322.5–607.5	0.027	not sig. in post hoc tests	
		Male study completers (*n* = 50)	480.0 ^a^	360.0–7.20.0	540.0 ^b^	442.5–660.0	450.0 ^a b^	300.0–600.0	0.001	0.009	0.003
		12-week adherers (*n* = 43)	480.0 ^a^	360.0–660.0	510.0 ^b^	420.0–660.0	420.0 ^a b^	300.0–600.0	0.001	0.007	0.002
		Female 12-week adherers (*n* = 19)	540.0	360.0–660.0	510.0	420.0–660.0	480.0	300.0–600.0	0.060		
		Male 12-week adherers (*n* = 24)	465.0	410.0–780.0	525.0 ^a^	427.5–682.5	360.0 ^a^	300.0–600.0	0.006	0.004	
		12-week dropouts (*n* = 38)	570.0 ^a^	345.0–720.0	592.5	360.0–675.0	480.0 ^a^	322.5–615.0	0.017	0.016	
		Non-starter (*n* = 11)	600.0	360.0–780.0	600.0	540.0–780.0	600.0	360.0–720.0	0.562		

Values marked ^a^ are significantly different (*p* value shown in column a), values marked ^b^ are significantly different (*p* value shown in column b).

**Table 7 ijerph-18-01700-t007:** Study completers: change in exercise self-efficacy.

Measure		Attendance Status	Baseline		12 Weeks		52 Weeks		Significance		
			Median	IQR	Median	IQR	Median	IQR	Overall	a	b
									α < 0.05	α < 0.0167	α < 0.0167
Self-Efficacy and Exercise Habits Survey	Making time for PA	All study completers (*n* = 92)	3.7 ^b^	3.0–4.3	3.7 ^a^	3.0–4.3	4.3 ^a b^	3.3–5.0	<0.0005	0.001	<0.0005
		Female study completers (*n* = 42)	3.2	2.7–3.8	3.4	2.8–4.3	3.7	2.7–4.4	0.642		
		Male study completers (*n* = 50)	3.9 ^a^	3.5–4.7	3.7 ^b^	3.0–4.4	5.0 ^a b^	4.3–5.0	<0.05	0.001	<0.0005
		12-week adherers (*n* = 43)	3.7	3.0–4.0	3.7 ^a^	3.0–4.3	4.3 ^a^	3.3–5.0	0.046	not sig. in post hoc tests	
		Female 12-week adherers (*n* = 19)	3.0	2.7–3.7	3.0	2.5–4.3	3.3	2.3–4.3	0.678		
		Male 12-week adherers (*n* = 24)	3.9 ^a^	3.5–4.7	3.7 ^b^	3.7–4.6	5.0 ^a b^	4.4–5.0	<0.0005	0.009	<0.0005
		12-week dropouts (*n* = 38)	3.7 ^a^	3.0–4.4	3.6 ^b^	2.8–4.3	4.5 ^a b^	3.7–5.0	0.001	0.014	<0.0005
		Non-starter (*n* = 11)	3.3	2.7–4.7	3.3	3.0–4.3	4.1	3.0–5.0	0.218		
	Sticking to PA	All study completers (*n* = 92)	3.7 ^a^	3.2–4.2	3.8 ^b^	3.2–4.3	4.5 ^a b^	3.5–5.0	<0.0005	<0.0005	<0.0005
		Female study completers (*n* = 42)	3.5 ^a^	3.0–4.3	3.8	3.0–4.3	4.0 ^a^	3.3–4.8	0.019	0.006	
		Male study completers (*n* = 50)	3.7 ^a^	3.3–4.3	3.6 ^b^	3.3–4.3	4.7 ^a b^	3.8–5.0	<0.0005	<0.0005	<0.0005
		12-week adherers (*n* = 43)	3.8 ^a^	3.3–4.0	4.0 ^b^	3.5–4.0	4.6 ^a b^	3.8–5.0	<0.0005	<0.0005	0.001
		Female 12-week adherers (*n* = 19)	3.7	3.0–4.2	3.8	3.4–4.5	4.0	3.5–4.8	0.171		
		Male 12-week adherers (*n* = 24)	3.9 ^a^	3.4–4.6	4.3 ^b^	3.5–4.8	4.8 ^a b^	4.5–5.1	0.001	0.004	0.005
		12-week dropouts (*n* = 38)	3.4 ^a^	3.1–4.1	3.6 ^b^	3.1–4.1	4.2 ^a b^	3.3–4.9	<0.0005	<0.0005	0.010
		Non-starter (*n* = 11)	3.4	2.5–3.8	3.0	2.8–3.8	3.6	3.3–5.0	0.202		

Values marked ^a^ are significantly different (*p* value shown in column a), values marked ^b^ are significantly different (*p* value shown in column b).

## Data Availability

The data presented in this study are available on request from the corresponding author. The data are not publicly available yet due to Edinburgh Napier University policy to make data available in a publically accessible repository six months after publication of results.

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
