# Peer review of "Gender Differences in Uptake, Adherence and Experiences: A Longitudinal, Mixed-Methods Study of a Physical Activity Referral Scheme in Scotland, UK"

_ijerph, 2021, doi:10.3390/ijerph18041700_

Round 1

Reviewer 1 Report

Dear Authors, 

Thank you for this interesting paper. Please, find some comments below. 

Please, note that I will only use line numbers because the page numbers are wrong for example after page 9, you have started again with page numbering at 2!

1- Validated Quantitative Questionnaires
Did they include two subjects, such as the research sample, in terms of gender, age, and the purpose of the questionnaire?
Meaning that it can refer to reliability and validity? 

2- Line 153 During week 12 and 52 interviews, we checked the accuracy of field notes by reflecting them back to participants during conversations, making corrections where necessary.

Can you  explain how did you ''reflecting them back to participants ''

3- For the results of themes and subthemes, I find it a little bit hard to follow or to compare the percentages which are the main aim, I would suggest presenting it in tables if possible so can the reader easily find the differences.

4- In the discussion section, please don't repeat everything from the result, just remind the reader and then try to interpret the results: When, why, what.

Reviewer 2 Report

Comments to author(s)

Thank you for submitting our manuscript titled ‘Gender differences in outcomes and experiences: a longitudinal, mixed methods study of a physical activity referral scheme in Scotland, UK.’ to IJERPH for consideration, which I read with great interest. This is a very well written manuscript in a very interesting and topical scientific/clinical field, and I applaud you for your mixed methods approach.

The whole paper is very well written and I only have minor comments.

Page 1, lines 14-19, something has gone wrong with your superscript numbers for the affiliations, as no2 seems to appear multiple times.

Line 54 change ‘is influenced’ with ‘appears to be influenced’ – to lower the tone of certainty of your argument

Data collection: where the telephone conversations recorded? And then transcribed? Please verify this and add to the manuscript along with a comment about anonymising the transcriptions.

Reviewer 3 Report

In examining gender differences in the uptake and adherence to PARS, alongside gendered perceptions of the scheme, this mixed methods study is both interesting and engaging. The authors address an ongoing, ambiguous topic within the literature and provide useful insights, that not only advances understanding of the area but will also be relevant to practitioners.

Overall, this is a well-considered and thought-out paper, that logically examines the current literature in the area and highlights those ambiguities that remain. The authors, to their credit, employ a longitudinal approach, still lacking in the PARS literature and do this with a generous qualitative sample, alongside the quantitative data. This is also achieved whilst representing a naturalistic setting, which again is valuable in demonstrating the ‘real world’ application of the authors’ findings. Whilst there are no specific aspects of the paper I would question; I did have some thoughts that the authors may want to consider. Within the title the word ‘outcomes’ was used and whilst the broad use of the term is relevant, in a PARS context this could imply ‘health outcomes’ rather than in relation to adherence and uptake. I wonder if something more indicative of the research focus would be appropriate. I also believe the authors demonstrate that the research advances understanding in the area of gender and PARS but I think there is opportunity to emphasise these novel contributions further, particularly within the discussion/ conclusion sections when making comparisons to previous literature. This could just be a subtle change in the language used but there is opportunity to really emphasise how these findings advance the current body of literature in this area.

Overall, though, the paper makes an original and valuable contribution to the available research literature, it flows well and effectively responds to the original aims, the authors are to be congratulated.

For the fairly minor issues, which caught my eye, and which the authors might want to address, there were only a few items I would point to:

P1 Line 29 – use of the word ‘starters’ could be considered a little vague and could be unclear for those less familiar with PARS, you use the phrase ‘those who started’ in reference to the scheme later in the paper, which reads more clearly.

P1 Line 35 – spelling of the word ‘individualisation’ but elsewhere in the paper American spelling is used.

P2 Line 56 – 59 This sentence currently lacks clarity and would benefit from rephrasing to articulate the point more clearly.

P2 Line 88 – I think it would be beneficial to name the validated questionnaires here rather than just the references.

P3 Line 101 – ‘ed group PA sessions’ appears in bold rather than normal font.

P3 Line 127/ 128 – It is detailed that the PARS provider supplied the research team with attendance data and yet HCPs sought written consent from the participant for only personal information (Line 112/112) to be released. To clarify was there an additional consent agreed with the PARS provider? Did this consent also include the release of data from the PARS provider. Would be useful to explain here.

P3 Line 131/ 132 – just to clarify, interviews were not recorded? Was there a particular reason for this? Were there limitations to only using field notes even when checking accuracy?

P15 Line 379 – the apostrophe is included in the sub-section numbers rather than before the title itself.

P15 Line 395 – there is no open brackets after the words ‘Participant 90’.

P16 Line 397 – there is repetition of the word ‘was’.

Round 2

Reviewer 1 Report

It looks like it has been modified.